# In-context Exploration-Exploitation for Reinforcement Learning

**Zhenwen Dai, Federico Tomasi, Sina Ghiassian**
Spotify Research
{zhenwend,federicot,sinag}@spotify.com

## Abstract

In-context learning is a promising approach for online policy learning of offline reinforcement learning (RL) methods, which can be achieved at inference time without gradient optimization. However, this method is hindered by significant computational costs resulting from the gathering of large training trajectory sets and the need to train large Transformer models. We address this challenge by introducing an In-context Exploration-Exploitation (ICEE) algorithm, designed to optimize the efficiency of in-context policy learning. Unlike existing models, ICEE performs an exploration-exploitation trade-off at inference time within a Transformer model, without the need for explicit Bayesian inference. Consequently, ICEE can solve Bayesian optimization problems as efficiently as Gaussian process biased methods do, but in significantly less time. Through experiments in grid world environments, we demonstrate that ICEE can learn to solve new RL tasks using only tens of episodes, marking a substantial improvement over the hundreds of episodes needed by the previous in-context learning method.

## 1 Introduction

Transformers represent a highly effective approach to sequence modelling, with applications extending across multiple domains like text, image, and audio. In the field of reinforcement learning (RL), Chen et al. (2021) and Janner et al. (2021) have suggested the concept of addressing offline RL as a sequential prediction problem using Transformer. This method has proven successful in tackling a range of tasks using solely offline training when combined with large-scale sequence modelling techniques (Lee et al., 2022; Reed et al., 2022a). A notable shortcoming lies with the policy's inability to self-improve when employed in online environments. To overcome this, fine-tuning methods have been introduced, like (Zheng et al., 2022), which enable continued policy improvement. However, these methods often rely on slow, computationally expensive gradient-based optimization.

Conversely, in-context learning, a characteristic observed in large language models (LLMs), can handle new tasks by providing the task details through language prompts, effectively eliminating the need for fine-tuning. Laskin et al. (2023) suggest an in-context learning algorithm for RL, which uses a sequence model to distill a policy learning algorithm from RL training trajectories. The resulting model is capable of conducting policy learning at inference-time through an iterative process of action sampling and prompt augmentation. This method incurs significant computational costs in collecting extensive training trajectory sets and training large Transformer models that are needed to model a substantial part of a training trajectory. The key reason for this high computational cost is the lengthy RL training trajectories resulting from the slow trial-and-error process of RL policy learning algorithms.

This paper aims to improve the efficiency of in-context policy learning by eliminating the need to learn from policy learning trajectories. In an ideal scenario, efficient policy learning could be achieved through an efficient trial-and-error process. For simplified RL problems such as multi-armed bandits (MAB), efficient trial-and-error process such as Thompson sampling and upper confidence bounds are proven to exist. This process, often referred to as the exploration-exploitation (EE) trade-off, relies heavily on the epistemic uncertainty derived from Bayesian belief. However, it is intractable to infer the exact epistemic uncertainty of sequential RL problems using conventional Bayesian methods. In light of recent studies on uncertainty estimation of LLMs (Yin et al., 2023),

we examine predictive distributions of sequence models, demonstrating that, by training with purely supervised learning on offline data, a sequence model can capture epistemic uncertainty in sequence prediction. This suggests the potential for implementing EE in offline RL.

Based on this observation, we develop an in-context exploration-exploitation (ICEE) algorithm for policy learning. ICEE takes as inputs a sequence of multiple episodes of the same task and predicts the corresponding action at every step conditioned on some hindsight information. This offline RL design resembles the Decision Transformer (DT, Chen et al., 2021), but ICEE tackles in-context policy learning by modeling multiple episodes of a task while DT only models a single episode. Furthermore, these episodes do not need to originate from a training trajectory, thereby averting the high computational costs associated with generating and consuming learning trajectories. The action distribution learned in DT is biased towards the data collection policy, which may not be ideal when it is sub-optimal. To address this bias, we introduce an unbiased objective and develop a specific form of hindsight information for efficient EE across episodes.

With experiments, we demonstrate that the EE behavior emerges in ICEE during inference thanks to epistemic uncertainty in action prediction. This is particularly evident when applying ICEE to Bayesian optimization (BO), as the performance of ICEE is on par with a Gaussian process based method on discrete BO tasks. We also illustrate that ICEE can successfully improve the policy for a new task with trials-and-errors from scratch for sequential RL problems. To the best of our knowledge, ICEE is the first method that successfully incorporates in-context exploration-exploitation into RL through offline sequential modelling.

## 2 RELATED WORK

**Meta Learning.** Interest in *meta learning* or *learning to learn* algorithms has recently increased. While learner is an agent that learns to solve a task using observed data, a learning to learn algorithm includes a *meta-learner* that continually improves the learning process of the learner (Schmidhuber et al., 1996; Thrun & Pratt, 2012; Hospedales et al., 2021; Sutton, 2022). Lots of work has been done within the space of meta learning. For example, Finn et al. (2017) proposed a general model-agnostic meta-learning algorithm that trains the model's initial parameters such that the model has maximal performance on a new task after the model parameters have been updated through a few gradient steps computed with a small amount of data from the new task. Other works on meta learning include improving optimizers (Andrychowicz et al., 2016; Li & Malik, 2016; Ravi & Larochelle, 2016; Wichrowska et al., 2017), improving few-shot learning (Mishra et al., 2017; Duan et al., 2017), learning to explore (Stadie et al., 2018), and unsupervised learning (Hsu et al., 2018).

In the space of deep meta RL (Wang et al., 2016), a few works focused on a special form of meta-learning called meta-gradients. In meta-gradients the meta-learner is trained by gradients by measuring the effect of meta-parameters on a learner that itself is also trained using a gradient algorithm (Xu et al., 2018). In other work, Zheng et al. (2018) used meta-gradients to learn rewards. Gupta et al. (2018) focused on automating the process of task design in RL, to free the expert from the burden of manual design of meta-learning tasks. Similarly, Veeriah et al. (2019) presented a method for a reinforcement learning agent to discover questions formulated as general value functions through the use of non-myopic meta-gradients. More recently, meta gradient reinforcement learning has seen substantial breakthroughs from performance gains on popular benchmarks to hybrid offline-online algorithms for meta RL (Xu et al., 2020; Zahavy et al., 2020; Flennerhag et al., 2021; Mitchell et al., 2021; Yin et al., 2023; Pong et al., 2022). The role of uncertainty in meta RL has been studied by Zintgraf et al. (2021), which result in an efficient online meta RL method. This work is then extended by Dorfman et al. (2021) to the off-policy setting.

**Offline Reinforcement Learning.** In general, reinforcement learning was proposed as a fundamentally online paradigm (Sutton, 1988; Sutton et al., 1999; Sutton & Barto, 2018). This online learning nature comes with some limitations such as making it difficult to adopt it to many applications for which it is impossible to gather online data and learn at the same time, such as autonomous driving and is also sometimes not as data efficient as it could bem since it might choose to learn from a sample and then dispose the sample and move on to the next (Levine et al., 2020). One idea for getting more out of the collected experience is to use replay buffers. When using buffers, part of the samples are kept in a memory and then are re-used multiple times so that the agent can learn more from them (Lin, 1992; Mnih et al., 2015). A variant of reinforcement learning, referred to as *offline RL* studies

RL algorithms that can learn fully offline, from a fixed dataset of previously gathered data without gathering new data at the time of learning (Ernst et al., 2005; Riedmiller, 2005; Lange et al., 2012; Fujimoto et al., 2019; Siegel et al., 2020; Gulcehre et al., 2020; Nair et al., 2020). Recent literature on decision transformers also focuses on offline RL (Chen et al., 2021) because it needs to calculate the *return to go* at training time, which in turn necessitates previously gathered data.

**In-context learning.** In-context RL algorithms are the ones that improve their policy entirely *in-context* without updating the network parameters or without any fine-tuning of the model(Lu et al., 2021). There has been some work studying the phenomenon of in-context learning trying to explain how learning in-context might be possible Abernethy et al. (2023); Min et al. (2022). The "Gato" agent developed by Reed et al. (2022b) works as a multi-model, multi-task, multi-embodiment generalist agent, meaning that the same trained agent can play Atari, caption images, chat, stack blocks with a real robot arm only based on its context. By training an RL agent at large scale, Team et al. (2023) showed that an in-context agent can adapt to new and open ended 3D environments. Of special interest to us is Algorithm Distillation (AD), which is a meta-RL method (Laskin et al., 2023). More specifically, AD is an in-context offline meta-RL method. Basically, AD is gradient-free—it adapts to downstream tasks without updating its network parameters.

## 3 EPISTEMIC UNCERTAINTY IN SEQUENCE MODEL PREDICTION

DT, also known as Upside-down RL, treats the offline policy learning problem as a sequence modelling problem. In this section, we consider a generic sequencing model and analyze its predictive uncertainty.

Let $\boldsymbol{X}_{1:T} = (\boldsymbol{x}_1, \ldots, \boldsymbol{x}_T)$ be a sequence of inputs of length $T$ and $\boldsymbol{Y}_{1:T} = (\boldsymbol{y}_1, \ldots, \boldsymbol{y}_T)$ be the corresponding sequence of outputs. Assume that the output sequence is generated following a step-wise probabilistic distribution parameterized by $\boldsymbol{\theta}$, $\boldsymbol{y}_t \sim p(\boldsymbol{y}_t|\boldsymbol{x}_t, \boldsymbol{X}_{1:t-1}, \boldsymbol{Y}_{1:t-1}, \boldsymbol{\theta})$. Each sequence is generated with a different parameter sampled from its prior distribution, $\boldsymbol{\theta} \sim p(\boldsymbol{\theta})$. These define a generative distribution of a sequence:

$$p(\boldsymbol{Y}_{1:T}, \boldsymbol{\theta}|\boldsymbol{X}_{1:T}) = p(\boldsymbol{\theta})p(\boldsymbol{y}_1|\boldsymbol{x}_1)\prod_{t=2}^{T} p(\boldsymbol{y}_t|\boldsymbol{x}_t, \boldsymbol{X}_{1:t-1}, \boldsymbol{Y}_{1:t-1}, \boldsymbol{\theta}). \tag{1}$$

A sequence modeling task is often defined as training an autoregressive model parameterized by $\boldsymbol{\psi}$, $p_{\boldsymbol{\psi}}(\boldsymbol{y}_t|\boldsymbol{x}_t, \boldsymbol{X}_{1:t-1}, \boldsymbol{Y}_{1:t-1})$, given a dataset of sequences $\mathcal{D} = \{\boldsymbol{X}^{(i)}, \boldsymbol{Y}^{(i)}\}_i$ generated from the above unknown generative distribution. At the limit of infinite data, the objective of maximum likelihood learning of the above sequence model can be formulated as $\boldsymbol{\psi}* = \arg\max_{\boldsymbol{\psi}} \mathcal{L}_{\boldsymbol{\psi}}$,

$$\mathcal{L}_{\boldsymbol{\psi}} = -\sum_t \int p(\boldsymbol{Y}_{1:t-1}|\boldsymbol{X}_{1:t-1})$$
$$D_{\mathrm{KL}}\left(p(\boldsymbol{y}_t|\boldsymbol{x}_t, \boldsymbol{X}_{1:t-1}, \boldsymbol{Y}_{1:t-1})||p_{\boldsymbol{\psi}}(\boldsymbol{y}_t|\boldsymbol{x}_t, \boldsymbol{X}_{1:t-1}, \boldsymbol{Y}_{1:t-1})\right) d\boldsymbol{Y}_{1:t-1} + C, \tag{2}$$

where $D_{\mathrm{KL}}(\cdot||\cdot)$ denotes the Kullback Leibler divergence and $C$ is a constant with respect to $\boldsymbol{\psi}$.

The left hand side distribution in the cross entropy term $p(\boldsymbol{y}_t|\boldsymbol{x}_t, \boldsymbol{X}_{1:t-1}, \boldsymbol{Y}_{1:t-1})$ is the *true* predictive distribution of $\boldsymbol{y}_t|\boldsymbol{x}_t$ conditioned on the observed history $\boldsymbol{Y}_{1:t-1}$ and $\boldsymbol{X}_{1:t-1}$, which can be written as

$$p(\boldsymbol{y}_t|\boldsymbol{x}_t, \boldsymbol{X}_{1:t-1}, \boldsymbol{Y}_{1:t-1}) = \int p(\boldsymbol{y}_t|\boldsymbol{x}_t, \boldsymbol{X}_{1:t-1}, \boldsymbol{Y}_{1:t-1}, \boldsymbol{\theta})p(\boldsymbol{\theta}|\boldsymbol{X}_{1:t-1}, \boldsymbol{Y}_{1:t-1})d\boldsymbol{\theta}, \tag{3}$$

where

$$p(\boldsymbol{\theta}|\boldsymbol{X}_{1:t-1}, \boldsymbol{Y}_{1:t-1}) = \frac{p(\boldsymbol{\theta})p(\boldsymbol{Y}_{1:t-1}|\boldsymbol{X}_{1:t-1}, \boldsymbol{\theta})}{\int p(\boldsymbol{\theta}')p(\boldsymbol{Y}_{1:t-1}|\boldsymbol{X}_{1:t-1}, \boldsymbol{\theta}')d\boldsymbol{\theta}'}. \tag{4}$$

As shown above, the *true* predictive distribution of $\boldsymbol{y}_t|\boldsymbol{x}_t$ contains both aleatoric uncertainty and epistemic uncertainty, in which the epistemic uncertainty is contributed by $p(\boldsymbol{\theta}|\boldsymbol{X}_{1:t-1}, \boldsymbol{Y}_{1:t-1})$. With sufficient data and model capacity, the generative distribution in the sequence model $p_{\boldsymbol{\psi}}(\boldsymbol{y}_t|\boldsymbol{x}_t, \boldsymbol{X}_{1:t-1}, \boldsymbol{Y}_{1:t-1})$ will be trained to match the *true* predictive distribution. As a result, we can expect epistemic uncertainty to be contained in the predictive distribution of a sequence model. Note that the predictive distribution can only capture the epistemic uncertainty with respect to the parameters of sequences $\boldsymbol{\theta}$, but does not include the epistemic uncertainty regarding the hyper-parameters (if exist).

## 4 IN-CONTEXT POLICY LEARNING

Epistemic uncertainty is the essential ingredient of EE. With the observation that the predictive distribution sequence model contains epistemic uncertainty, we design an in-context policy learning algorithm with EE.

Consider the problem of solving a family of RL games based on offline data. From each game, a set of trajectories are collected from a number of policies, where $\tau_k^{(i)} = (\boldsymbol{o}_{k,1}^{(i)}, \boldsymbol{a}_{k,1}^{(i)}, r_{k,1}^{(i)}, \ldots, \boldsymbol{o}_{k,T_k}^{(i)}, \boldsymbol{a}_{k,T_k}^{(i)}, r_{k,T_k}^{(i)})$ is the trajectory of the $k$-th episode of the $i$-th game and $\boldsymbol{o}, \boldsymbol{a}, r$ denote the observed state, action and reward respectively. The policy used to collect $\tau_k^{(i)}$ is denoted as $\pi_k^{(i)}(\boldsymbol{a}_{k,t}^{(i)}|\boldsymbol{o}_{k,t}^{(i)})$. We concatenate all the episodes of the $i$-th game into a single sequence $\boldsymbol{\tau}^{(i)} = (\tau_1^{(i)}, \ldots, \tau_K^{(i)})$. For convenience, the superscript $^{(i)}$ will be omitted in the following text unless the $i$-th game is explicitly referred to.

We propose a sequence model that is trained to step-wisely predict $p_{\boldsymbol{\psi}}(\boldsymbol{a}_{k,t}|R_{k,t}, \boldsymbol{o}_{k,t}, \boldsymbol{H}_{k,t})$, where $R_{k,t}$ is the *return-to-go* at the $k$-th episode and the $t$ time step and $\boldsymbol{H}_{k,t} = (\tau_{k,1:t-1}, \boldsymbol{\tau}_{1:k-1})$ is the history until the time step $t$ including the past episodes[1]. The above model formulation is similar to DT but a sequence in DT only contains one episode. Note that, different from AD, the concatenated trajectories do not need to be from a RL learning algorithm.

As shown in the previous section, by doing maximum likelihood learning on the collected trajectories, the predictive distribution will be trained to match the *true* posterior distribution of action of the data collection policy,

$$p(\boldsymbol{a}_{k,t}|R_{k,t}, \boldsymbol{o}_{k,t}, \boldsymbol{H}_{k,t}) = \frac{p(R_{k,t}|\boldsymbol{a}_{k,t}, \boldsymbol{o}_{k,t}, \boldsymbol{H}_{k,t})\pi_k(\boldsymbol{a}_{k,t}|\boldsymbol{o}_{k,t})}{\int p(R_{k,t}|\boldsymbol{a}'_{k,t}, \boldsymbol{o}_{k,t}, \boldsymbol{H}_{k,t})\pi_k(\boldsymbol{a}'_{k,t}|\boldsymbol{o}_{k,t})d\boldsymbol{a}'_{k,t}}, \tag{5}$$

where $p(R_{k,t}|\boldsymbol{a}_{k,t}, \boldsymbol{o}_{k,t}, \boldsymbol{H}_{k,t})$ is the distribution of return after the time step $t$ following $\pi_k$.

As shown in (5), the posterior distribution of action is biased towards the data collection policy. Following such an action distribution allows us to reproduce the trajectories generated by the data collection policy but will lead to recreation of suboptimal trajectories if the data collection policy is not optimal. A more desirable action distribution is the action distribution that corresponds to the specified return without the influence of data collection policy, i.e.

$$\hat{p}(\boldsymbol{a}_{k,t}|R_{k,t}, \boldsymbol{o}_{k,t}, \boldsymbol{H}_{k,t}) = \frac{p(R_{k,t}|\boldsymbol{a}_{k,t}, \boldsymbol{o}_{k,t}, \boldsymbol{H}_{k,t})\mathcal{U}(\boldsymbol{a}_{k,t})}{\int p(R_{k,t}|\boldsymbol{a}'_{k,t}, \boldsymbol{o}_{k,t}, \boldsymbol{H}_{k,t})\mathcal{U}(\boldsymbol{a}'_{k,t})d\boldsymbol{a}'_{k,t}}, \tag{6}$$

where $\mathcal{U}(\boldsymbol{a}_{k,t})$ is the uniform random policy, which gives all the actions equal probabilities. To let the sequence model learn the unbiased action distribution, the maximum likelihood objective needs to be defined as

$$\mathcal{L}_{\boldsymbol{\psi}} = \sum_{k,t} \int \hat{p}(R_{k,t}, \boldsymbol{a}_{k,t}|\boldsymbol{o}_{k,t}, \boldsymbol{H}_{k,t}) \log p_{\boldsymbol{\psi}}(\boldsymbol{a}_{k,t}|R_{k,t}, \boldsymbol{o}_{k,t}, \boldsymbol{H}_{k,t})dR_{k,t}d\boldsymbol{a}_{k,t}. \tag{7}$$

After applying the importance sampling trick, the Monte Carlo approximation of the above objective can be derived as

$$\mathcal{L}_{\boldsymbol{\psi}} \approx \sum_{k,t} \frac{\mathcal{U}(\boldsymbol{a}_{k,t})}{\pi_k(\boldsymbol{a}_{k,t}|\boldsymbol{o}_{k,t})} \log p_{\boldsymbol{\psi}}(\boldsymbol{a}_{k,t}|R_{k,t}, \boldsymbol{o}_{k,t}, \boldsymbol{H}_{k,t}), \tag{8}$$

where $\boldsymbol{a}_{k,t} \sim \pi_k(\boldsymbol{a}_{k,t}|\boldsymbol{o}_{k,t})$ and $R_{k,t} \sim p(R_{k,t}|\boldsymbol{a}_{k,t}, \boldsymbol{o}_{k,t}, \boldsymbol{H}_{k,t})$, i.e., $\boldsymbol{a}_{k,t}$ and $R_{k,t}$ are sampled from the data collection policy $\pi_k$.

## 5 DESIGN OF RETURN-TO-GO

Return-to-go is a crucial component of DT for solving RL tasks at inference using a trained sequence model. Its return-to-go scheme is designed to calculate the expected return signal from a single episode. In order to achieve in-context policy learning, we design the return-to-go across episodes.

---

[1] Although the learned action distribution does not need to condition on the return-to-go of the past actions, for the convenience of a causal Transformer implementation, $\boldsymbol{H}_{k,t}$ contains the return-to-go of the past actions.

---

**Algorithm 1:** In-context Exploration-Exploitation (ICEE) Action Inference

---

**Input:** A trained ICEE model $p_\psi$,
$\boldsymbol{H}_{1,1} = \{\}$ ;
**for** *each episode* $k = 1, \ldots, K$ **do**
    Reset the environment;
    Select the episode return-to-go $\tilde{c}_k$ ;
    **for** *each step* $t = 1, \ldots, T_k$ **do**
        Get an observation $\boldsymbol{o}_{k,t}$ ;
        Sample the step return-to-go $c_{k,t} \sim q(c_{k,t})$ ;
        Sample an action $\boldsymbol{a}_{k,t} \sim p_\psi(\boldsymbol{a}_{k,t}|R_{k,t}, \boldsymbol{o}_{k,t}, \boldsymbol{H}_{k,t})$ ;
        Take the sampled action $\boldsymbol{a}_{k,t}$ and collect the reward $r_{k,t}$;
        Expand the history $\boldsymbol{H}_{k,t+1} = \boldsymbol{H}_{k,t} \cup \{R_{k,t}, \boldsymbol{o}_{k,t}, \boldsymbol{a}_{k,t}, r_{k,t}\}$;
    **end**
    Compute the true return-to-go based on the whole episode $\{\hat{R}_{k,t}\}_{t=1}^{T_k}$;
    Update the history $\boldsymbol{H}_{k,T_k}$ with the true return-to-go $\{\hat{R}_{k,t}\}_{t=1}^{T_k}$;
**end**

---

The return-to-go of ICEE consists of two components: one for the individual steps within an episode and the other for the cross-episode behavior, $R_{k,t} = (c_{k,t}, \tilde{c}_k)$. The in-episode return-to-go $c_{k,t}$ follows the design used in (Chen et al., 2021), which is defined as the cumulative rewards starting from the current step $c_{k,t} = \sum_{t'>t} r_{k,t'}$. This design borrows the concept of the cumulative reward of RL and has the benefit of encapsulating the information of the future rewards following the policy. This is very useful when the outcomes of future steps strongly depend on the state and action of the current step. It allows the action that leads to a good future outcome to be differentiated from the action that leads to a bad future outcome in the sequence model. The downside is that with a non-expert data collection policy, the optimal return-to-go at each state is often unobserved. This will limits the ability of the sequence model to achieve better performance than the data collection policy at inference time.

In the cross-episode return-to-go design, the situation is different. The initial states of individual episodes are independent of each other. What determines the cumulative rewards of individual episodes are the sequence of actions. If we consider the whole policy space as the action space for each episode, the cross-episode decision making is closer to MAB, where the policy is the action and the return of an episode is the MAB reward. Driven by this observation, we define the return-to-go based on the improvement of the return of current episode compared to all the previous episodes. Specifically, we define the cross-episode return-to-go as

$$\tilde{c}_k = \begin{cases} 1 & \bar{r}_k > \max_{1 \le j \le k-1} \bar{r}_j, \\ 0 & \text{otherwise.} \end{cases} \tag{9}$$

where $\bar{r}_k = \sum_t r_{k,t}$ is the cumulative reward of the $k$-th episode. Intuitively, at inference time, by conditioning on $\tilde{c}_k = 1$, we take actions from a policy that is "sampled" according to the probability of being better than all the previous episodes. This encourages the sequence model to deliver better performance after collecting more and more episodes. This design avoids the limitation of the need of observing the optimal policy learning trajectories.

**Action Inference.** After training the sequence model, the model can be used to perform policy learning from scratch. At each step, we sample an action from the sequence model conditioned on the trajectory so far and a return-to-go for the step. The return-to-go for action sampling is defined as follows. The cross-episode return-to-go $\tilde{c}_k$ is always set to one to encourage policy improvements. For the in-episode return-to-go, we follow the action inference proposed by Lee et al. (2022). During training of ICEE, a separate sequence model is trained to predict the discretized return from the trajectories, $p_\phi(c_{k,t}|\tilde{c}_k, \boldsymbol{o}_{k,t}, \boldsymbol{H}_{k,t})$. At inference time, an in-episode return-to-go for each step is sampled from an augmented distribution

$$q(c_{k,t}) \propto p_\phi(c_{k,t}|\tilde{c}_k, \boldsymbol{o}_{k,t}, \boldsymbol{H}_{k,t})\left(\frac{c_{k,t} - c_{\min}}{c_{\max} - c_{\min}}\right)^\kappa. \tag{10}$$

This augmentation biases the return-to-go distribution towards higher values, which encourages the agent to take actions that leads to better returns. The return prediction is not combined into the main

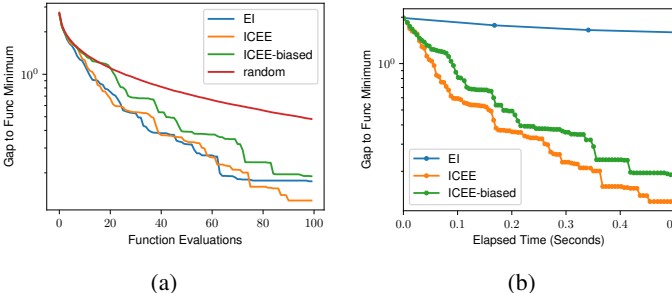

Figure 1: Discrete BO results on 2D functions with 1024 candidate locations. 16 benchmark functions are used with five trials each. The y-axis shows the average distance between the current best estimate and the true minimum. The x-axis of (a) is the number of function evaluations and the x-axis of (b) is the elapsed time.

sequence model as in (Lee et al., 2022), because in the main sequence model the returns are also fed into the inputs. In this way, the model quickly figures out $c_{k,t}$ can be predicted from $c_{k,t-1}$. This is problematic because at inference time the true return cannot be observed until the end of an episode.

After $c_{k,t}$ is sampled, an action is sampled conditioned on the combined return-to-go $R_{k,t}$. The resulting state and reward are concatenated to the trajectory for the next step action prediction. At the end of an episode, we will recalculate the true $c_{k,t}$ and $\tilde{c}_k$ based on the rewards from the whole episode and update the return-to-go in the trajectory with the recalculated ones. This makes the trajectory at inference time be as close to the training trajectories as possible. The description of the action inference algorithm can be found in Algorithm 1.

## 6 BAYESIAN OPTIMIZATION EXPERIMENTS

Bayesian optimization (BO) is a very successful application of exploration-exploitation (EE). It is able to search for the optimum of a function with the minimum number of function evaluations. The Gaussian process (GP) based BO methods have been widely used in various domains like hyperparameter tuning, drug discovery, aerodynamic optimization. To evaluate the EE performance of ICEE, we apply it to BO and compare it with a GP-based approach using one of most widely used acquisition functions, expected improvement (EI).

We consider a discrete BO problem. The task is to find the location from a fixed set of points that has the lowest function value as few number of function evaluations as possible. BO can be viewed as a special type of multi-armed bandit (MAB), where each action is associated with a location in a bounded space. To solve BO with ICEE, we encode the iterative search trajectory of a function as a single sequence, where $a_t$ is the location of which a function value is collected at the step $t$, $r_t$ is the corresponding function value and $R_t$ is the return-to-go. The observations $\{o_t\}$ can be used to encode any side information that is known about the function and the minimum. As no such information is available for generic BO, we do not use $\{o_t\}$ in our experiment. As the function value can seen as the available immediate reward for, we treat each action as a different episode and use only the episode return-to-go as the $R_t$ here. As each action is associated with a location, we embed actions by learning a linear projection between the location space and the embedding space. When decoding a Transformer output for an action, the logit of an action is generated using a MLP that takes as input the Transformer output together with the embedding associated with the action. The design is to tackle the challenge that the set of locations of each function may be different.

To train ICEE to solve the discrete BO problem, we need to generate training data that consists of input-output pairs of randomly sampled functions. During training, the input-output pairs at random locations are generated on-the-fly. We use a GP with the Matérn 5/2 kernel to sample 1024 points for each function. The locations of these points are sampled from a uniform distribution on $[0, 1]$. The length scales of the kernel are sampled from a uniform distribution on $[0.05, 0.3]$.

Ideally, for each sampled function, we can use a BO algorithm to solve the search problem and take the sequence of actions as the training data. This is very computationally expensive due to

the high computational cost of BO algorithms. Thanks to the EE property of ICEE, the training data do not need to come from an expert inference algorithm. This is very different from behavior cloning / algorithm distillation. We use a cheap data collection algorithm that randomly picks actions according to a probability distribution that is computed based on the function values of actions. The probability distribution is defined as follows: The actions are sorted in an ascending order according to their function values and the action at the $i$-th position gets the probability $p(\boldsymbol{a} = \boldsymbol{a}_i) \propto \exp(\frac{1024-i}{1023}\gamma)$, $\gamma$ is sampled from a uniform distribution between 0 and 10 for each function. Intuitively, the locations with lower function values are more likely to be picked.

After training ICEE, we evaluate its performance on a set of 2D benchmark functions. We use 16 2D functions implemented in (Kim & Choi, 2017). The input space of every function is normalized to be between 0 and 1. We randomly sample a different set of 1024 points from each function and normalize the resulting function values to be zero-mean and unit-variance. Each function was given five trials with different initial designs. At each step of search, we compute the difference of the function value between the current best estimate and the true minimum. The average performance of all the evaluated functions is shown in Fig. 1. The performance of ICEE is compared with the GP-based BO method using the EI acquisition function and the random baseline that picks locations according to a uniform random probability. "ICEE-biased" denotes a variant of ICEE that does not use the action bias correction objective as shown in (8). The search efficiency of ICEE is on par with the GP-based BO method with EI. Both of them are significantly better than random and are noticeably better than ICEE-biased. The performance gap between ICEE and ICEE-biased shows the loss of efficiency due to the biased learned action distribution. Being able to perform on par with a state-of-the-art BO method demonstrates that ICEE is able to perform state-of-the-art EE with in-context inference.

A clear advantage of ICEE is that the whole search is done through model inference without need of any gradient optimization. In contrast, GP-based BO methods need to fit a GP surrogate function at each step, which results into a significant speed difference. Fig. 1b shows the same search results with the x-axis being the elapsed time. All the methods run on a single A100 GPU. Thanks to the in-context inference, ICEE is magnitude faster than conventional BO methods. More details of the BO experiments can be found in Appendix D.

## 7 RL EXPERIMENTS

We investigate the in-context policy learning capability of ICEE on sequential RL problems. To demonstrate the capability of in-context learning, we focus on the families of environments that cannot be solved through zero-shot generalization of a pre-trained model, so in-context policy learning is necessary for solving the tasks. This implies that some information that are important to the success of a task is missing from the state representation and will need to be discovered by the agent. We use the two grid world environments in (Lee et al., 2022): dark room and dark key-to-door. The details of them are as follows.

**Dark Room.** The experiment takes place in a 2D discrete POMDP, where an agent is placed inside a room to locate a goal spot. The agent has access to its own location coordinates $(x, y)$, but is unaware of the goal's placement requiring it to deduce it from the rewards received. The room's dimensions are 9x9 with the possible actions by the agent including moving one step either left, right, up or down, or staying idle, all within an episode length of 20. Upon completion, the agent gets placed back at the mid-point of the map. Two environment variants are considered for this experiment: The Dark Room case where the agent obtains a reward (r=1) each time the goal is achieved, and the Dark Room Hard case where the rewards are sparse (r=1 only once for attaining the goal). Whenever the reward value is not 1, it will be considered as 0. Different from (Lee et al., 2022), we keep the room size of the hard case to be 9 x 9.

**Dark Key-to-Door.** This setting is similar to the Dark Room, but with added challenging features. The task of the agent is to locate an invisible key to receive a one-time reward of r=1, and subsequently, identify an invisible door to gain another one-time reward of r=1. Otherwise, the reward remains at r=0. The agent's initial location in each episode resets randomly. The room size is still 9 x 9 but the episode length increases to 50 steps.

To collect data for offline training, we sample a set of new games for each mini-batch. We collect $K$ episodes from each game. Thanks to the EE capability of ICEE, the training data do not need to be from a real RL learning algorithm like Deep Q-Network (DQN), which is expensive to run. Instead, we let a cheap data collection policy act for $K$ episodes independently and concatenate the resulting episodes into a single sequence. We use an $\epsilon$-greedy version of the "cheating" optimal policy. The policy knows the goal location that is unknown to the agent and will move straightly towards the goal with $1 - \epsilon$ probability and with $\epsilon$ probability it will take an action that does not make the agent closer to the goal. For each episode, $\epsilon$ is sampled from a uniform distribution between 0 and 1. Intuitively, this policy has some chance to solve a game efficiently when $\epsilon$ is small but on average it does not deliver a good performance. For Dark Room experiments, each sequence consists of 50 episodes and for Dark Key-To-Door, it consists of 20 episodes.

**Dark Room (Biased).** To demonstrate the benefits of EE for sequential RL problem when the data collection policy cannot be optimal, we create a variant of the dark room environment. At each step, the data collection policy takes the "left" action with the $2/3$ probability and with the $1/3$ probability acts as described above. At training time, the goal can be at any where in the room and, at evaluation time, the goal will only appear on the right hand side where $x > 5$.

For sequential RL problems, ICEE consists of two sequence models: one for action prediction and the other for the in-episode return-to-go prediction. The return-to-go sequence model takes as inputs the sequence of state, action, reward triplets and predicts the in-episode return-to-go. The action prediction model takes as inputs the sequence of the triplets and the two return-to-go $R_{k,t}$ and predicts the action sequence. The two models are trained together with the same gradient optimizer. To encourage ICEE to solve the games quickly, when calculating the in-episode return-to-go, a negative reward, $-1/T$, is given to each step that does not receive a reward, where $T$ is the episode length. Both $\tilde{c}_k$ and $c_{k,t}$ are discrete and tokenized.

## 7.1 BASELINE METHODS

**Source.** We use the data collection policy as a baseline for comparison. As the data collection policy solves each episode independently, we calculate the average return across multiple episodes.

**Algorithm Distillation (AD, Laskin et al., 2023).** The in-context learning algorithm that distills a RL algorithms from RL training trajectories. AD predicts action based only on the current states and the history of state, action and reward triplets. We replicate the implementation of AD using the Transformer architecture as ICEE. We apply AD to the same training data as ICEE uses (ignoring the return-to-go signals), despite they are generated from RL learning trajectories.

**AD-sorted.** AD is designed to be trained on RL learning trajectories. An important property of RL learning trajectories is that the performance of the agent gradually increases throughout training. To mimic such trajectories using our data, we sort the episodes in a sequence according to the sampled $\epsilon$ of the data collection policy in the descent order. $\epsilon$ determines how close the data collection policy is to the optimal policy. In this order, the episodes in a latter position of sequence tend to have a higher return. We train AD using this sorted sequences instead of the original ones.

**Multi-game Decision Transformer (MGDT, Lee et al., 2022).** MGDT is not an in-context learning algorithm. We train MGDT using only one episode from each sampled game. The performance of MGDT shows that what the performance of the agent is when there is no in-context policy learning.

## 7.2 EVALUATION & RESULTS

After training, ICEE will be evaluated on solving a set of sampled games. The inference algorithm is described in Alg. 1. No online model update is performed by ICEE and all the baseline methods at evaluation time. For each sampled game, ICEE and two variants of AD will act for $K$ episodes consecutively. In each episode, the trajectories from the past episodes are used as in the history representation. Ideally, a good performing agent identifies the missing information with as few number of episodes as possible and then maximizes the return in the following episodes. For each problem, we sample 100 games and $K$ is 50 for Dark Room and 20 for Key-To-Door.

The experiment results are shown in Fig. 2. ICEE is able to solve the sampled games efficiently compared to the baseline methods. The EE capability allows ICEE to search for the missing infor-

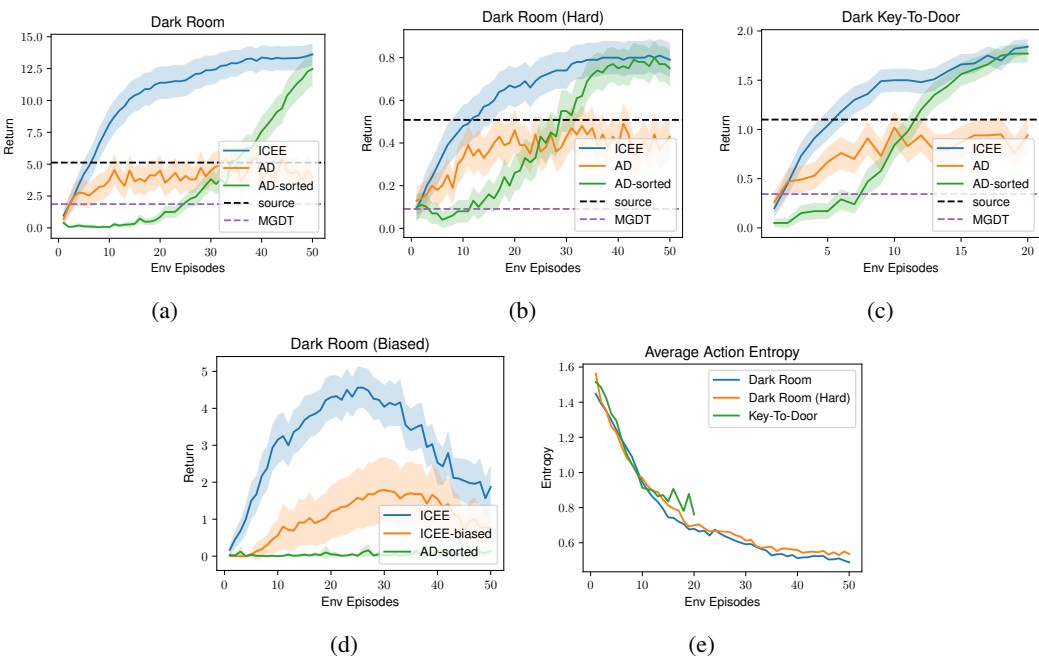

Figure 2: Experimental results of in-context policy learning on grid world RL problems. An agent is expected to solve a game by interacting with the environment for $K$ episodes without online model updates. The y-axis of (a-d) shows the average returns over 100 sampled games after each episode. The y-axis of (e) shows the entropy of the action distribution. The x-axis shows the number of episodes that an agent experiences.

mation efficiently and then acts with confidence once the missing information is found. An indicator of such behavior is the continuous decrease of the action entropies as the agent experiences more episodes (see Fig. 2e).

As expected, the original AD learns to mimic the data collection policy, which results in an average performance slightly below the data collection policy. MGDT fails to solve most of the games due to the missing information. Interestingly, despite that the training data is not generated from a RL learning algorithm, AD-sorted is able to clone the behavior of the data collection policy with different $\epsilon$ at different stages, which allows it to solve the games at the end of the sequence.

ICEE-biased is not shown in Fig. 2a, Fig. 2b and Fig. 2c as it achieves similar performance as ICEE does. The reason is that there is no clear bias in the action distribution of the data collection policy. However, as shown in Fig. 2d, for the Dark Room (Biased) environment, ICEE clearly outperforms ICEE-biased, as it can overcome the bias in the data collection policy and keep sufficient uncertainty in action distribution to explore the right hand side of the room. AD-sorted fails the task because it clones the data collection policy, which is unlikely to solve the tasks due to the action bias.

## 8 CONCLUSION

In this paper, we analyze the predictive distribution of sequence models and show that the predictive distribution can contain epistemic uncertainty, which inspires the creation of an EE algorithm. We present an in-context EE algorithm by extending the DT formulation to in-context policy learning and deriving an unbiased training objective. Through the experiments on BO and discrete RL problems, we demonstrate that: (i) ICEE can perform EE in in-context learning without the need of explicit Bayesian inference; (ii) The performance of ICEE is on par with state-of-the-art BO methods without the need of gradient optimization, which leads to significant speed-up; (iii) New RL tasks can be solved within tens of episodes.

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

# Appendix

## A   DERIVATION OF THE OBJECTIVE OF A SEQUENCE MODEL

The maximum likelihood objective of a sequence model in (2) can be derived in the following steps.

$$
\begin{aligned}
\mathcal{L}_{\boldsymbol{\psi}} &= \int p(\boldsymbol{Y}_{1:T}, \boldsymbol{\theta}|\boldsymbol{X}_{1:T}) \log p_{\boldsymbol{\psi}}(\boldsymbol{Y}_{1:T}|\boldsymbol{X}_{1:T}) d\boldsymbol{Y}_{1:T} d\boldsymbol{\theta} \\
&= \int p(\boldsymbol{Y}_{1:T}, \boldsymbol{\theta}|\boldsymbol{X}_{1:T}) \log \prod_{t=1}^{T} p_{\boldsymbol{\psi}}(\boldsymbol{y}_t|\boldsymbol{x}_t, \boldsymbol{X}_{1:t-1}, \boldsymbol{Y}_{1:t-1}) d\boldsymbol{Y}_{1:T} d\boldsymbol{\theta} \\
&= \sum_{t=1}^{T} \int p(\boldsymbol{Y}_{1:T}, \boldsymbol{\theta}|\boldsymbol{X}_{1:T}) \log p_{\boldsymbol{\psi}}(\boldsymbol{y}_t|\boldsymbol{x}_t, \boldsymbol{X}_{1:t-1}, \boldsymbol{Y}_{1:t-1}) d\boldsymbol{Y}_{1:T} d\boldsymbol{\theta} \\
&= \sum_{t=1}^{T} \int p(\boldsymbol{Y}_{1:t}|\boldsymbol{X}_{1:t}) \log p_{\boldsymbol{\psi}}(\boldsymbol{y}_t|\boldsymbol{x}_t, \boldsymbol{X}_{1:t-1}, \boldsymbol{Y}_{1:t-1}) d\boldsymbol{Y}_{1:t} \\
&= \sum_{t=1}^{T} \int p(\boldsymbol{Y}_{1:t-1}|\boldsymbol{X}_{1:t-1}) p(\boldsymbol{y}_t|\boldsymbol{x}_t, \boldsymbol{X}_{1:t-1}, \boldsymbol{Y}_{1:t-1}) \log p_{\boldsymbol{\psi}}(\boldsymbol{y}_t|\boldsymbol{x}_t, \boldsymbol{X}_{1:t-1}, \boldsymbol{Y}_{1:t-1}) d\boldsymbol{Y}_{1:t} \\
&= \sum_{t=1}^{T} \int p(\boldsymbol{Y}_{1:t-1}|\boldsymbol{X}_{1:t-1}) \Big( p(\boldsymbol{y}_t|\boldsymbol{x}_t, \boldsymbol{X}_{1:t-1}, \boldsymbol{Y}_{1:t-1}) \log p_{\boldsymbol{\psi}}(\boldsymbol{y}_t|\boldsymbol{x}_t, \boldsymbol{X}_{1:t-1}, \boldsymbol{Y}_{1:t-1}) \\
&\quad - p(\boldsymbol{y}_t|\boldsymbol{x}_t, \boldsymbol{X}_{1:t-1}, \boldsymbol{Y}_{1:t-1}) \log p(\boldsymbol{y}_t|\boldsymbol{x}_t, \boldsymbol{X}_{1:t-1}, \boldsymbol{Y}_{1:t-1}) \Big) d\boldsymbol{Y}_{1:t} \\
&\quad + \int p(\boldsymbol{Y}_{1:t-1}|\boldsymbol{X}_{1:t-1}) p(\boldsymbol{y}_t|\boldsymbol{x}_t, \boldsymbol{X}_{1:t-1}, \boldsymbol{Y}_{1:t-1}) \log p(\boldsymbol{y}_t|\boldsymbol{x}_t, \boldsymbol{X}_{1:t-1}, \boldsymbol{Y}_{1:t-1}) d\boldsymbol{Y}_{1:t} \\
&= \sum_{t=1}^{T} \int p(\boldsymbol{Y}_{1:t-1}|\boldsymbol{X}_{1:t-1}) \Big( - D_{\mathrm{KL}}\left(p(\boldsymbol{y}_t|\boldsymbol{x}_t, \boldsymbol{X}_{1:t-1}, \boldsymbol{Y}_{1:t-1}) || p_{\boldsymbol{\psi}}(\boldsymbol{y}_t|\boldsymbol{x}_t, \boldsymbol{X}_{1:t-1}, \boldsymbol{Y}_{1:t-1})\right) \Big) d\boldsymbol{Y}_{1:t-1} \\
&\quad + \int p(\boldsymbol{Y}_{1:t-1}|\boldsymbol{X}_{1:t-1}) H\Big( p(\boldsymbol{y}_t|\boldsymbol{x}_t, \boldsymbol{X}_{1:t-1}, \boldsymbol{Y}_{1:t-1}) \Big) d\boldsymbol{Y}_{1:t-1} \\
&= - \sum_{t} \int p(\boldsymbol{Y}_{1:t-1}|\boldsymbol{X}_{1:t-1}) D_{\mathrm{KL}}\left(p(\boldsymbol{y}_t|\boldsymbol{x}_t, \boldsymbol{X}_{1:t-1}, \boldsymbol{Y}_{1:t-1}) || p_{\boldsymbol{\psi}}(\boldsymbol{y}_t|\boldsymbol{x}_t, \boldsymbol{X}_{1:t-1}, \boldsymbol{Y}_{1:t-1})\right) d\boldsymbol{Y}_{1:t-1} + C,
\end{aligned}
$$

## B   DERIVATION OF THE UNBIASED OBJECTIVE

The Decision Transformer's approach to offline RL includes the training of an action distribution conditioned the return, thereby allowing for the sampling of an action at the time of inference by providing the projected return (return-to-go). Given that the return is the result of present and downstream actions, the distribution of action that a model tries to learn can be reframed as an action's posterior distribution, as is presented in equation (5). Note that equation (5) outlines the data distribution, which should be distinguished from the neural network model $p_{\boldsymbol{\psi}}(\boldsymbol{a}_{k,t}|R_{k,t}, \boldsymbol{o}_{k,t}, \boldsymbol{H}_{k,t})$. As noted in equation (5), the action's distribution is proportionate to the return's distribution, and this is subsequently weighted by the probability of action from the data collection policy. As it is the posterior distribution derived by the Bayes rule, we denote it as the "true" action posterior distribution.

To put this intuitively, if a model can accurately match equation (5), it will result in the action distribution being skewed towards the data collection policy. For instance, in a pre-recorded sequence, should an action be randomly selected with extremely low probability from the data collection policy, yet yield a high return, the subsequent action distribution in equation (5) would attribute a minute probability to the relevant action, given the high return. Despite the observation of a high return following the action, which would suggest a high probability of $p(R_{k,t}|\boldsymbol{a}_{k,t}, \boldsymbol{o}_{k,t}, \boldsymbol{H}_{k,t})$, the resulting probability of action is weighted by the probability of action in the data collection policy,

$\pi_k(\boldsymbol{a}_{k,t}|\boldsymbol{o}_{k,t})$, resulting in a small value. Therefore, even though (5) is the true posterior distribution of action, it is not the desirable action distribution for our model.

Ideally, the action distribution, as demonstrated in equation (6), should be proportional only to the return's distribution and would be unaffected by the data collection policy. With such a distribution, the undesired devaluation due to the data collection policy would be eliminated, thereby resolving the mentioned issue.

As explained above, we would like to learn an action distribution in (6) instead of the action distribution in (5). However, since (5) is the true action distribution of data, the common maximum likelihood training objective will let the model match the action distribution in (5).

The unbiased objective of learning the action distribution in (8) can be derived in the following steps.

To let the model learn the action distribution in (6) instead, we first state the desirable training objective as if the data follow the distribution (6):

$$
\mathcal{L}_{\boldsymbol{\psi}} = \sum_{k,t} \int \hat{p}(R_{k,t}, \boldsymbol{a}_{k,t}|\boldsymbol{o}_{k,t}, \boldsymbol{H}_{k,t}) \log p_{\boldsymbol{\psi}}(\boldsymbol{a}_{k,t}|R_{k,t}, \boldsymbol{o}_{k,t}, \boldsymbol{H}_{k,t}) dR_{k,t} d\boldsymbol{a}_{k,t}
$$

$$
= \sum_{k,t} \int \hat{p}(R_{k,t}|\boldsymbol{o}_{k,t}, \boldsymbol{H}_{k,t}) \Big( \int \hat{p}(\boldsymbol{a}_{k,t}|R_{k,t}, \boldsymbol{o}_{k,t}, \boldsymbol{H}_{k,t}) \log p_{\boldsymbol{\psi}}(\boldsymbol{a}_{k,t}|R_{k,t}, \boldsymbol{o}_{k,t}, \boldsymbol{H}_{k,t}) d\boldsymbol{a}_{k,t} \Big) dR_{k,t}
$$

Then, we apply the importance sampling trick to introduce the true action posterior in the equation:

$$
\mathcal{L}_{\boldsymbol{\psi}} = \sum_{k,t} \int \hat{p}(R_{k,t}|\boldsymbol{o}_{k,t}, \boldsymbol{H}_{k,t}) \Big( \int p(\boldsymbol{a}_{k,t}|R_{k,t}, \boldsymbol{o}_{k,t}, \boldsymbol{H}_{k,t}) \frac{\hat{p}(\boldsymbol{a}_{k,t}|R_{k,t}, \boldsymbol{o}_{k,t}, \boldsymbol{H}_{k,t})}{p(\boldsymbol{a}_{k,t}|R_{k,t}, \boldsymbol{o}_{k,t}, \boldsymbol{H}_{k,t})}
$$
$$
\log p_{\boldsymbol{\psi}}(\boldsymbol{a}_{k,t}|R_{k,t}, \boldsymbol{o}_{k,t}, \boldsymbol{H}_{k,t}) d\boldsymbol{a}_{k,t} \Big) dR_{k,t}
$$

After some rearrangement of the equation, we get a much clearer formulation of the objective:

$$
\mathcal{L}_{\boldsymbol{\psi}} = \sum_{k,t} \int \hat{p}(R_{k,t}|\boldsymbol{o}_{k,t}, \boldsymbol{H}_{k,t}) \Big( \int p(\boldsymbol{a}_{k,t}|R_{k,t}, \boldsymbol{o}_{k,t}, \boldsymbol{H}_{k,t}) \frac{\frac{p(R_{k,t}|\boldsymbol{a}_{k,t}, \boldsymbol{o}_{k,t}, \boldsymbol{H}_{k,t}) \mathcal{U}(\boldsymbol{a}_{k,t})}{\hat{p}(R_{k,t}|\boldsymbol{o}_{k,t}, \boldsymbol{H}_{k,t})}}{\frac{p(R_{k,t}|\boldsymbol{a}_{k,t}, \boldsymbol{o}_{k,t}, \boldsymbol{H}_{k,t}) \pi_k(\boldsymbol{a}_{k,t}|\boldsymbol{o}_{k,t})}{p(R_{k,t}|\boldsymbol{o}_{k,t}, \boldsymbol{H}_{k,t})}}
$$
$$
\log p_{\boldsymbol{\psi}}(\boldsymbol{a}_{k,t}|R_{k,t}, \boldsymbol{o}_{k,t}, \boldsymbol{H}_{k,t}) d\boldsymbol{a}_{k,t} \Big) dR_{k,t}
$$
$$
= \sum_{k,t} \int \hat{p}(R_{k,t}|\boldsymbol{o}_{k,t}, \boldsymbol{H}_{k,t}) \Big( \int p(\boldsymbol{a}_{k,t}|R_{k,t}, \boldsymbol{o}_{k,t}, \boldsymbol{H}_{k,t}) \frac{\mathcal{U}(\boldsymbol{a}_{k,t}) p(R_{k,t}|\boldsymbol{o}_{k,t}, \boldsymbol{H}_{k,t})}{\pi_k(\boldsymbol{a}_{k,t}|\boldsymbol{o}_{k,t}) \hat{p}(R_{k,t}|\boldsymbol{o}_{k,t}, \boldsymbol{H}_{k,t})}
$$
$$
\log p_{\boldsymbol{\psi}}(\boldsymbol{a}_{k,t}|R_{k,t}, \boldsymbol{o}_{k,t}, \boldsymbol{H}_{k,t}) d\boldsymbol{a}_{k,t} \Big) dR_{k,t}
$$
$$
= \sum_{k,t} \int p(R_{k,t}|\boldsymbol{o}_{k,t}, \boldsymbol{H}_{k,t}) \Big( \int p(\boldsymbol{a}_{k,t}|R_{k,t}, \boldsymbol{o}_{k,t}, \boldsymbol{H}_{k,t}) \frac{\mathcal{U}(\boldsymbol{a}_{k,t})}{\pi_k(\boldsymbol{a}_{k,t}|\boldsymbol{o}_{k,t})}
$$
$$
\log p_{\boldsymbol{\psi}}(\boldsymbol{a}_{k,t}|R_{k,t}, \boldsymbol{o}_{k,t}, \boldsymbol{H}_{k,t}) d\boldsymbol{a}_{k,t} \Big) dR_{k,t}
$$
$$
= \sum_{k,t} \int p(R_{k,t}, \boldsymbol{a}_{k,t}|\boldsymbol{o}_{k,t}, \boldsymbol{H}_{k,t}) \frac{\mathcal{U}(\boldsymbol{a}_{k,t})}{\pi_k(\boldsymbol{a}_{k,t}|\boldsymbol{o}_{k,t})} \log p_{\boldsymbol{\psi}}(\boldsymbol{a}_{k,t}|R_{k,t}, \boldsymbol{o}_{k,t}, \boldsymbol{H}_{k,t}) d\boldsymbol{a}_{k,t} dR_{k,t}
$$

Note the probability distribution in front is now the joint distribution of return and action in the data distribution. We apply the Monte Carlo approximation to the integral by considering the recorded data are samples from the data distribution. We get the proposed training objective.

$$
\mathcal{L}_{\boldsymbol{\psi}} \approx \sum_{k,t} \frac{\mathcal{U}(\boldsymbol{a}_{k,t})}{\pi_k(\boldsymbol{a}_{k,t}|\boldsymbol{o}_{k,t})} \log p_{\boldsymbol{\psi}}(\boldsymbol{a}_{k,t}|R_{k,t}, \boldsymbol{o}_{k,t}, \boldsymbol{H}_{k,t}), \quad R_{k,t}, \boldsymbol{a}_{k,t} \sim p(R_{k,t}, \boldsymbol{a}_{k,t}|\boldsymbol{o}_{k,t}, \boldsymbol{H}_{k,t})
$$

## C    IMPLEMENTATION DETAILS

ICEE is implemented based on nanoGPT [2]. For the RL experiments, ICEE contains 12 layers with 128 dimensional embeddings. There are 4 heads in the multi-head attention. We use the Adam optimizer with the learning rate $10^{-5}$.

## D    BAYESIAN OPTIMIZATION EXPERIMENTS

We consider a discrete BO problem. The task is to find the location from a fixed set of points that has the lowest function value as few number of function evaluations as possible. At the beginning of search, the function values of some randomly picked locations are given. The BO algorithm will be asked to suggest a location where the function minimum lies and then, the function value will be collected from the suggested location. The suggestion-evaluation loop will be repeated with a fixed number of times. The performance of a BO algorithm is evaluated by how quickly it can find the function minimum.

The list of 2D functions used for evaluations are: Branin, Beale, Bohachevsky, Bukin6, DeJong5, DropWave, Eggholder, GoldsteinPrice, HolderTable, Kim1, Kim2, Kim3, Michalewicz, Shubert, SixHumpCamel, ThreeHumpCamel.

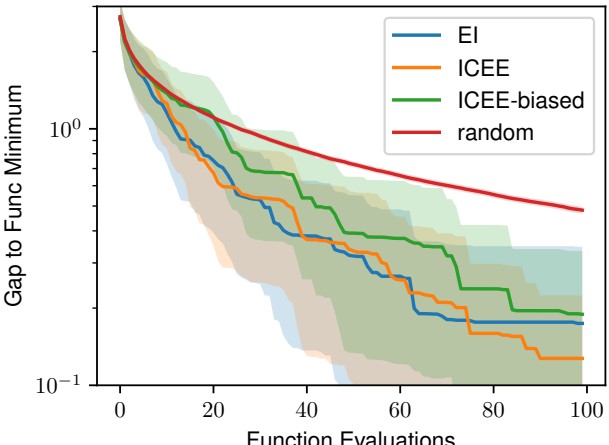

Figure 3: Discrete BO results on 2D functions with 1024 candidate locations. The shading area shows the 95% confidence interval of the mean.

The expected improvement baseline is implemented using BOTorch (Balandat et al., 2020). We used the "SingleTaskGP" class for the GP surrogate model, which uses a Matern 5/2 kernel with a Gamma prior over the length scales.

## E    QUANTITATIVE RESULTS OF THE DISCRETE RL EXPERIMENTS

Please find the quantitative comparison of the RL experiments shown in Fig. 2 below. The shown values are the average returns over 100 sampled games and the values in the parentheses are the confidence intervals of the mean estimates, which correspond to the shaded area in the figure. We take three time points along the trajectories of in-context policy learning. As MGDT is not able to update policy at inference time, we only estimate a single mean return for each game.

---

[2]https://github.com/karpathy/nanoGPT

|  | Dark Room (10th Episode) | Dark Room (30th Episode) | Dark Room (50th Episode) |
|---|---|---|---|
| ICEE | 8.15 (1.29) | 12.37 (1.14) | 13.61 (0.86) |
| AD | 3.74 (1.15) | 4.51 (1.17) | 4.03 (1.15) |
| AD-sorted | 0.05 (0.05) | 3.83 (0.87) | 12.48 (1.37) |
| MGDT | 1.86 (0.93) | 1.86 (0.93) | 1.86 (0.93) |
| source | 5.13 (1.19) | 5.13 (1.19) | 5.13 (1.19) |

|  | Dark Room (Hard) (10th) | Dark Room (Hard) (30th) | Dark Room (Hard) (50th) |
|---|---|---|---|
| ICEE | 0.48 (0.10) | 0.74 (0.09) | 0.79 (0.08) |
| AD | 0.33 (0.09) | 0.43 (0.10) | 0.43 (0.10) |
| AD-sorted | 0.08 (0.05) | 0.55 (0.10) | 0.75 (0.08) |
| MGDT | 0.09 (0.06) | 0.09 (0.06) | 0.09 (0.06) |
| source | 0.51 (0.10) | 0.51 (0.10) | 0.51 (0.10) |

|  | Dark Key-To-Door (5th) | Dark Key-To-Door (10th) | Dark Key-To-Door (20th) |
|---|---|---|---|
| ICEE | 1.04 (0.15) | 1.50 (0.12) | 1.84 (0.08) |
| AD | 0.67 (0.15) | 1.02 (0.17) | 0.94 (0.17) |
| AD-sorted | 0.17 (0.08) | 0.84 (0.14) | 1.77 (0.09) |
| MGDT | 0.34 (0.11) | 0.34 (0.11) | 0.34 (0.11) |
| source | 1.10 (0.19) | 1.10 (0.19) | 1.10 (0.19) |

