# OpenReview forum: "In-context Exploration-Exploitation for Reinforcement Learning"
_ICLR.cc/2024/Conference — ICLR 2024 poster_

### Official Review · Reviewer_cBRg · 2023-10-30

**Soundness:** 3 good
**Presentation:** 3 good
**Contribution:** 4 excellent
**Rating:** 8
**Confidence:** 4

**Summary:**

The paper presents a novel algorithm called In-context Exploration-Exploitation (ICEE). Learning to learn in context is promising but demands extensive learning trajectory data and expansive training of large Transformer models. The ICEE algorithm is designed to address these challenges by augmenting decision transformers with an extra cross-episode return-to-go and training on concatenated episodic trajectories. It learns to balance exploration versus exploitation directly within the Transformer model during inference, eliminating the need for separate Bayesian inference processes. This enables ICEE to tackle Bayesian optimization tasks as effectively as methods based on Gaussian processes but at a notably faster rate. Experimental results show that ICEE can adeptly learn and solve new RL tasks with just tens of episodes, corroborating its efficiency.

**Strengths:**

**Learning from Episodic Trajectories**: ICEE learns in-context reinforcement learning using merely concatenated episodic trajectories, which makes it a versatile choice for scenarios where abundant learning trajectories are unavailable and constitutes a novel and significant contribution to field of reinforcement learning.

**Connection to Thompson Sampling**: ICEE resolves the exploration-exploitation dilemma in a way analogous to posterior sampling methods, like Thompson Sampling. Given the history, it samples a move that could possibly lead to an improved outcome, gaining either a high return or new information. This method could be the most scalable and efficient implementation among all Thompson Sampling variants.

**Weaknesses:**

**Ambiguous Presentation**: Section 3 of the paper feels disjointed from its succeeding sections, leading to some confusion regarding the role and definition of the parameter $\theta$ within ICEE's framework.

**Questionable Experimental Setup**: The experiments predominantly utilize noisy optimal policy variants for data collection. This leaves an open question regarding ICEE's performance when relying on trajectories from random policies.

**Questions:**

1. How is the exploration-exploitation dilemma addressed in ICEE?
2. How does ICEE perform when trained with data collected by random policies?

I will be pleased to raise my score if these questions, especially the first one, can be properly answered.

---

> ### Author Response · Authors · 2023-11-20
> **Re: Official Review of Submission1828 by Reviewer cBRg**
>
> We thank the reviewer for the feedback.
>
> > How is the exploration-exploitation dilemma addressed in ICEE?
>
> The exploration-exploitation dilemma in ICEE is primarily addressed through a method analogous to a posterior sampling style of exploration-exploitation, such as Thompson Sampling. The specific algorithm, however, depends on the return-to-go design.
> For instance, applying our cross-episode return-to-go to the Bayesian optimization problem, sampling actions from our learned model equates to sampling an action from the renowned probability of improvement acquisition function. This is what allows our model to deliver performance comparable to Bayesian Optimization with an expected improvement acquisition function.
>
> Section 3 sheds more light on this dilemma by demonstrating that a generic sequence model's predictive distribution exhibits epistemic uncertainty if the model is trained with sufficient data and possesses enough model capacity. Applying this to the in-context policy learning problem, we consider a specific sequence model that predicts an action based on return-to-go, current state, and observed history: $p(a_{k,t} | R_{k,t}, o_{k,t}, H_{k,t})$. Using the notation from equation (3), $y_t$ corresponds to $a_{k,t}$, $x_t$ corresponds to $R_{k,t}$ and $o_{k,t}$, and $X_{1:t-1}$ corresponds to $H_{k,t}$, excluding past actions.
>
> Equation (3) illustrates that the action distribution obtained from data is the marginalization of the task's posterior distribution given the observed history. Thus, sampling an action from the action distribution $p(a_{k,t} | R_{k,t}, o_{k,t}, H_{k,t})$ is equivalent to first sampling a task based on the task's posterior distribution $p(\theta | H_{k,t})$, then sampling an action conditioned on the sampled task $p(a_{k,t} | R_{k,t}, o_{k,t}, H_{k,t}, \theta)$. The latter is an explicit posterior sampling algorithm. This equivalence explains why our model can effectively balance the exploration-exploitation trade-off.
>
> To illustrate this point further, consider the Bayesian optimization problem. Here, an action corresponds to an input location from which the unknown function's value can be taken. $\theta$ is the parameterization of the unknown function, the minimum of which we are seeking. One way to apply Thompson Sampling to BO is to use a two-step sampling process. For a scenario where there are two observed input-output pairs of the function $H_3 = (x_1, y_1, x_2, y_2)$, we can derive the posterior $p(\theta | H_3)$ of the unknown function. A specific parameterization is then sampled from the posterior $\theta’ \sim p(\theta | H_3)$. Based on $\theta'$, an action distribution is derived, proportional to the likelihood of each input location having a lower function value than both $y_1$ and $y_2$. This can be represented as $p(a_3 | R_3=true, H_3, \theta’)$. An action is then sampled from this distribution, equivalent to sampling according to the probability of improvement. This action — applied through the Thompson sampling algorithm — can enable the discovery of the function's minimum. The aforementioned relationship demonstrate that the sampling of an action from our model is equivalent to the two-step sampling process, and so, our model can also detect the minimum of the unknown function, as substantiated by our experimental results.
>
> > How does ICEE perform when trained with data collected by random policies?
>
> ICEE can learn a good policy from data collected using random policies. In data collected using a full random policy, the chance of having good behaviors are lower. It means that ICEE will probably need more data and have a longer training time. We will include a small scale experiment to demonstrate this behavior.
>
>
> > Benchmark Implementation Concerns: The study's application of Algorithm Distillation (AD) diverges from the original practice, focusing on cloning a static behavior policy instead of true learning trajectories.
>
> Our proposed method, ICEE, isn't designed to be an in-context behavior cloning algorithm and the experiments conducted do not attempt to establish ICEE complicity as a superior in-context behavior cloning algorithm to AD. Our intent was to demonstrate the advantages of employing in-context exploration-exploitation compared to in-context behavior cloning, using AD as our benchmark for the latter.
>
> An enlightening secondary finding from our paper is that AD can learn to execute in-context policy learning from non-learning trajectory data, provided the episodes are organized according to performance. This outcome can be attributed to the epistemic uncertainty present in the action distribution, which results in naturally occurring in-context exploration-exploitation behavior.

---

> > ### Comment · Reviewer_cBRg · 2023-11-20
> >
> > I appreciate the authors' comprehensive rebuttal, which has effectively addressed my major concerns. The connection between ICEE and Thompson sampling is both convincing and illuminating. Given the significance of this connection, I recommend incorporating this discussion into the main body of the paper for the benefit of readers. This addition will undoubtedly enrich the paper's content and provide clearer insights into the methodology. In light of these revisions and clarifications, I am pleased to revise my evaluation to a score of 8 (Good Paper).

---

### Official Review · Reviewer_5MSL · 2023-10-31

**Soundness:** 4 excellent
**Presentation:** 3 good
**Contribution:** 3 good
**Rating:** 8
**Confidence:** 3

**Summary:**

The authors develop a in-context exploration-exploitation algorithm for reinforcement learning. This algorithm can be viewed as an extension and refinement of Decision Transformer by generalizing the model from a single episode to multiple episode, and modified the objective to eliminate the bias. The algorithm is extensively tested against Bayesian optimization problems and multiple reinforcement
learning tasks, the proposed algorithm is able to solve sampled games much faster than baseline methods.

**Strengths:**

The experimental results are very promising. While the algorithm is an extension of Decision Transformer(CT), it certainly outperforms DT significantly.

The application of in-context learning in RL is innovative and has potentials to bring advancement to learning.

**Weaknesses:**

The explanation provided in the paper on why the proposed algorithm is not very convincing. There is no quantitative analysis provided for identifying the difference between ICEE and DT. As pointed below, some of the explanations of the key ideas in the paper are not very clear.

**Questions:**

On page 4, below eq (5), the statement "true posterior distribution of action is biased towards the data collection policy" is not that straightforward from eq (5), could you add more explanations?

Between eq.(7) and eq.(8), maybe the author could be provide some details on the importance sampling trick they mentioned.

---

> ### Author Response · Authors · 2023-11-20
> **Re: Official Review of Submission1828 by Reviewer 5MSL (1)**
>
> We thank the reviewer for the feedback.
>
> >  As pointed below, some of the explanations of the key ideas in the paper are not very clear. On page 4, below eq (5), the statement "true posterior distribution of action is biased towards the data collection policy" is not that straightforward from eq (5), could you add more explanations?
>
> The Decision Transformer's approach to offline RL includes the training of an action distribution conditioned the return, thereby allowing for the sampling of an action at the time of inference by providing the projected return (return-to-go). Given that the return is the result of present and downstream actions, the distribution of action that a model tries to learn can be reframed as an action's posterior distribution, as is presented in equation (5). Note that equation (5) outlines the data distribution, which should be distinguished from the neural network model $p_{\psi}(a_{k,t} | R_{k,t}, o_{k,t}, H_{k,t})$. As noted in equation (5), the action's distribution is proportionate to the return's distribution, and this is subsequently weighted by the probability of action from the data collection policy. As it is the posterior distribution derived by the Bayes rule, we denote it as the “true” action posterior distribution.
>
> To put this intuitively, if a model can accurately match equation (5), it will result in the action distribution being skewed towards the data collection policy. For instance, in a pre-recorded sequence, should an action be randomly selected with extremely low probability from the data collection policy, yet yield a high return, the subsequent action distribution in equation (5) would attribute a minute probability to the relevant action, given the high return. Despite the observation of a high return following the action, which would suggest a high probability of $p(R_{k,t}| a_{k,t}, o_{k,t}, H_{k,t})$, the resulting probability of action is weighted by the probability of action in the data collection policy, $\pi_k(a_{k,t}|o_{k,t})$, resulting in a small value. Therefore, even though (5) is the true posterior distribution of action, it is not the desirable action distribution for our model.
>
> Ideally, the action distribution, as demonstrated in equation (6), should be proportional only to the return's distribution and would be unaffected by the data collection policy. With such a distribution, the undesired devaluation due to the data collection policy would be eliminated, thereby resolving the mentioned issue.
>
> > Between eq.(7) and eq.(8), maybe the author could provide some details on the importance sampling trick they mentioned.
>
> As explained in the previous answer, we would like to learn an action distribution in (6) instead of the action distribution in (5). However, since (5) is the true action distribution of data, the common maximum likelihood training objective will let the model match the action distribution in (5).
>
> To let the model learn the action distribution in (6) instead, we first state the desirable training objective as if the data follow the distribution (6):
> $$L_{\psi} = \sum_{k,t} \int \hat{p}(R_{k,t}, a_{k,t} |o_{k,t}, H_{k,t}) \log p_{\psi}(a_{k,t} | R_{k,t}, o_{k,t}, H_{k,t}) dR_{k,t} d a_{k,t}$$
>
> Then, we apply the importance sampling trick to introduce the true action posterior in the equation:
> $$L_{\psi} = \sum_{k,t} \int\hat{p}(R_{k,t} |o_{k,t}, H_{k,t}) \Big( \int p(a_{k,t} |R_{k,t}, o_{k,t}, H_{k,t}) \frac{\hat{p}(a_{k,t} |R_{k,t}, o_{k,t}, H_{k,t})}{p(a_{k,t} |R_{k,t}, o_{k,t}, H_{k,t})}  \log p_{\psi}(a_{k,t} | R_{k,t}, o_{k,t}, H_{k,t})  d a_{k,t} \Big) dR_{k,t}$$
>
> After some rearrangement of the equation, we get a much clearer formulation of the objective (the detailed steps can be found in appendix):
> $$\sum_{k,t}  \int p(R_{k,t}, a_{k,t} | o_{k,t}, H_{k,t}) \frac{\mathcal{U}(a_{k,t})}{\pi_k(a_{k,t}|o_{k,t})}  \log p_{\psi}(a_{k,t} | R_{k,t}, o_{k,t}, H_{k,t})  d a_{k,t} dR_{k,t}$$
>
> Note the probability distribution in front is now the joint distribution of return and action in the data distribution. We apply the Monte Carlo approximation to the integral by considering the recorded data are samples from the data distribution. We get the proposed training objective.
> $$ \sum_{k,t} \frac{\mathcal{U}(a_{k,t})}{\pi_k(a_{k,t}|o_{k,t})}  \log p_{\psi}(a_{k,t} | R_{k,t}, o_{k,t}, H_{k,t}), \quad R_{k,t}, a_{k,t} \sim p(R_{k,t}, a_{k,t} | o_{k,t}, H_{k,t}) $$

---

> > ### Author Response · Authors · 2023-11-20
> > **Re: Official Review of Submission1828 by Reviewer 5MSL (2)**
> >
> > > There is no quantitative analysis provided for identifying the difference between ICEE and DT.
> >
> > Please find the quantitative comparison of the RL experiments shown in Figure 2 below. The shown values are the average returns over 100 sampled games and the values in the parentheses are the confidence intervals of the mean estimates, which correspond to the shaded area in the figure. We take three time points along the trajectories of in-context policy learning. As MGDT is not able to update policy at inference time, we only estimate a single mean return for each game.
> >
> > | | Dark Room (10th Episode) | Dark Room (30th Episode) | Dark Room (50th Episode) |
> > | -------- | ------- | ------- | ------- |
> > |ICEE |8.15 (1.29) |12.37 (1.14) |13.61 (0.86) |
> > |AD |3.74 (1.15) |4.51 (1.17) |4.03 (1.15) |
> > |AD-sorted |0.05 (0.05) |3.83 (0.87) |12.48 (1.37) |
> > |MGDT|1.86 (0.93)|1.86 (0.93)|1.86 (0.93)|
> > |source|5.13 (1.19)|5.13 (1.19)|5.13 (1.19)|
> >
> > | | Dark Room (Hard) (10th Episode) | Dark Room (Hard) (30th Episode) | Dark Room (Hard) (50th Episode) |
> > | -------- | ------- | ------- | ------- |
> > |ICEE |0.48 (0.10) |0.74 (0.09) |0.79 (0.08) |
> > |AD |0.33 (0.09) |0.43 (0.10) |0.43 (0.10) |
> > |AD-sorted |0.08 (0.05) |0.55 (0.10) |0.75 (0.08) |
> > |MGDT|0.09 (0.06)|0.09 (0.06)|0.09 (0.06)|
> > |source|0.51 (0.10)|0.51 (0.10)|0.51 (0.10)|
> >
> > | | Dark Key-To-Door (5th Episode) | Dark Key-To-Door (10th Episode) | Dark Key-To-Door (20th Episode) |
> > | -------- | ------- | ------- | ------- |
> > |ICEE |1.04 (0.15) |1.50 (0.12) |1.84 (0.08) |
> > |AD |0.67 (0.15) |1.02 (0.17) |0.94 (0.17) |
> > |AD-sorted |0.17 (0.08) |0.84 (0.14) |1.77 (0.09) |
> > |MGDT|0.34 (0.11)|0.34 (0.11)|0.34 (0.11)|
> > |source|1.10 (0.19)|1.10 (0.19)|1.10 (0.19)|

---

> > > ### Comment · Reviewer_5MSL · 2023-11-22
> > >
> > > Thank you for providing detailed responses. They address my concerns to some degree, I have updated my score.

---

### Official Review · Reviewer_1NXH · 2023-11-01

**Soundness:** 3 good
**Presentation:** 3 good
**Contribution:** 2 fair
**Rating:** 5
**Confidence:** 3

**Summary:**

This paper studies in-context RL and proposes a method that addresses the exploration-exploitation tradeoff. Specifically, during the pre-training, the method assigns a reward to each trajectory so that the model can learn to improve the policy. Furthermore, the paper applies the method on discrete Bayesian Optimization and several RL environments, which show better performance than those of some baselines such Algorithm Distillation.

**Strengths:**

+ In-context RL is a timely topic in the study of RL and this work provides some interesting idea about the design of in-context RL algorithms.
+ The design of cross-episode reward successfully removes the requirement that the offline dataset is generated from some RL learning algorithms. This design improves the applicability of in-context RL.
+ The experimental results show promising performance of ICEE in the early episodes.

**Weaknesses:**

- The experimental evaluation is not sufficient. There is no comparison between in-context RL algorithms and traditional offline RL algorithms, multi-task RL algorithms or posterior sampling based algorithms.
- The proposed ICEE algorithm lacks theoretical performance guarantees.
- The ICEE algorithm heavily relies on importance sampling and Monte Carlo approximation, which are widely used techniques in traditional RL and not new.

**Questions:**

1.	The return-to-go for cross-episode seems simple and straightforward. Is it possible to have a more sophisticated design such that the performance can be further improved? For example, current return-to-go seems depends on the order of the offline trajectories. Is there any possible way to avoid this and make design to be order-insensitive?

---

> ### Author Response · Authors · 2023-11-19
> **Re: Official Review of Submission1828 by Reviewer 1NXH**
>
> We thank the reviewer for the feedback.
>
> > The ICEE algorithm heavily relies on importance sampling and Monte Carlo approximation, which are widely used techniques in traditional RL and not new.
>
> While it is true that Importance Sampling and Monte Carlo approximations are common in probabilistic ML, the implementation and application in the ICEE algorithm are done in a unique and novel manner. To the best of our knowledge, the bias in action distribution learning has not previously been explored extensively in RL via supervised learning literature. Our approach, using an importance sampling-based solution, demonstrates an effective reduction in bias in the action distribution, as evidenced in the experimental section. Therefore, we firmly believe that even though these techniques are common, their use in this context does not detract from the overall contribution of the paper.
>
>
> > The return-to-go for cross-episode seems simple and straightforward. Is it possible to have a more sophisticated design such that the performance can be further improved? For example, current return-to-go seems depends on the order of the offline trajectories. Is there any possible way to avoid this and make design to be order-insensitive?
>
> Our cross-episode return-to-go is computed according to the sequence of episodes in the offline data. However, this does not bind it to a specific episode order. In our experimental setup, episodes were gathered independently and arranged randomly in order of creation. ICEE's ability to learn effectively from randomly ordered episodes indicates that the return-to-go design is indeed robust against episode sequence. The performance disparity observed between AD and AD-sorted provides further evidence that episode order-sensitive methods, like AD, cannot perform well in this scenario.
>
> Our cross-episode return-to-go method generates binary values consistent across tasks. This is a clear advantage over the discounted cumulative reward design in Decision Transformer (DT). Cumulative rewards can fluctuate significantly between tasks. Multi-game Decision Transformer (MGDT) addresses this by training a specific model to predict individual sequence cumulative rewards. During inference, a bias is added to the predicted distribution to use as return-to-go. However, this technique often fails in the problems where expected cumulative rewards are hard to predict such as Bayesian Optimization (BO). In BO, a basic cumulative reward design is the distance between the function value at the current location and the function value's minimum over the sequence. This return-to-go design would not work well for BO, because predicting a function’s minimum is a notoriously hard problem. Our cross-episode return-to-go avoids this issue by generating consistent values across tasks. For instance, in the Bandit setting, the return-to-go is calculated during training by comparing each action's reward with previous ones.  During inference, this action probability of the model corresponds to the probability of improving the reward when setting the return-to-go to "true". For BO, our return-to-go equates to the well-known probability of improvement acquisition function in BO literature.
>
> > The experimental evaluation is not sufficient. There is no comparison between in-context RL algorithms and traditional offline RL algorithms, multi-task RL algorithms or posterior sampling based algorithms.
>
> This paper primarily focuses on understanding the epistemic uncertainty in sequence model prediction and developing an in-context exploration exploitation algorithm. In our experiments, ICEE successfully performed well on exploration-exploitation in BO and also proved to be more efficient at solving 2D discrete RL problems than Algorithm Distillation. The paper does not claim any comparisons of ICEE's performance against traditional offline RL algorithms. Moreover, conventional offline RL methodologies would either struggle with policy learning at inference time due to the task family's POMDP setting or need online updates like RL$^2$. Such methods cannot be directly compared to ICEE for the same reason why online and offline RL methods are not comparable.

---

> > ### Comment · Reviewer_1NXH · 2023-11-22
> >
> > Thank you for the response! After going through the authors' responses to all reviewers and the paper again, I am not convinced that the contribution of this work is very significant, considering that importance sampling is a common technique to address distribution shift in RL, and the design of return-to-of is not principled. Therefore I decide to keep my original rating.

---

### Official Review · Reviewer_5QaL · 2023-11-05

**Soundness:** 3 good
**Presentation:** 4 excellent
**Contribution:** 3 good
**Rating:** 8
**Confidence:** 4

**Summary:**

This paper proposes a novel way to leverage the powerful autoregressive sequence learning capabilities in transformer architectures to decision making problems by finding a way to bring in exploration-exploitation for in context policy learning. This goes beyond recent work along with a similar theme for decision making problems (Decision transformers, Algorithm distillation) in two important ways. First, it learns to explore and exploit across multiple episodes concatenated as one sequence, and secondly it is designed to exploit data generated from arbitrary policies that can, in principle, be highly inefficient.

Experiments are conducted on two well motivated settings -- (1) bayesian optimization and (2) a family of episodic toy RL tasks in a 2d discrete maze. The authors also investigate controlled variations in the data collection policy to highlight how the proposed technique compares favorably to Algorithm Distillation.

**Strengths:**

- Simple and well articulated but technically precise characterization of the issues surrounding in context policy learning, especially related to epistemic uncertainty.
- The experimental results, while in simple domains at small scale, are nevertheless well motivated and designed to illustrate the promise of the proposed ideas.
- For the BO benchmark, the proposed approach is competitive with EI, but at a small fraction of the cost. For the sequential RL tasks, it achieves better performance compared to other comparable meta learning approaches which were not designed to do exploration in context.

**Weaknesses:**

The context length requires a sequence of episodes for in-context learning, which can make it fundamentally quite challenging in terms of scale to go beyond small dimensional problems.

Nits/minor typos:
- GP biased -> GP based
- pg 3: indefinite -> infinite
- Sec 6: wildly -> widely

**Questions:**

- The last statement in page 3 seems somewhat ambiguous, could the authors clarify what they mean by this? e.g. _"...Note that the epistemic uncertainty associated with cross sequence latent variables if exist, e.g., the posterior of parameters in $p(\theta)$, would be not learned into the sequence model."_ But shouldn't the true predictive distribution of $y_t$ implicitly encode both the epistemic and aleatoric in the limit of infinite data?

- There could be several reasonable ways to generate suboptimal data for analysis in toy experiments, and the authors have investigated an $\epsilon-$greedy version of the (oracle) optimal where $\epsilon$ is sampled independently for each _episode_ with full $[0,1]$ support. As long as there are enough samples covering a large enough $\epsilon$, a non-trivial part of the episode collection would effectively be oracle policy demos? If that's the case, a more effective stress might be more meaningful e.g. as one possible minimal change, one could sample $\epsilon$ more generally as a beta distribution with two params and rerun the experiments across a range of those params.

---

> ### Author Response · Authors · 2023-11-14
> **Re: Official Review of Submission1828 by Reviewer 5QaL**
>
> We thank the reviewer's feedback. We will fix the typos.
>
> > The context length requires a sequence of episodes for in-context learning, which can make it fundamentally quite challenging in terms of scale to go beyond small dimensional problems.
>
> We acknowledge that the sequence length required for our in-context policy learning indeed exceeds what needed for the Decision Transformer. However, in contrast to the preceding work, Algorithm distillation, it is already  significantly shorter. In applications such as Bayesian optimization, where the sequence typically involves hundreds of evaluations, our proposed algorithm is directly applicable to real-world problems. Thanks to the continuous advancements in large language models (LLMs), the prompt size that can be processed is rapidly increasing. For example, the most recent GPT-4 Turbo can handle 128K tokens. With such improvement, we could envision that in-context policy learning can become applicable to real-world RL problems.
>
> > The last statement in page 3 seems somewhat ambiguous, ...
>
> Indeed, the predictive distribution encapsulates both the epistemic and aleatoric uncertainty in the context of limitless data. The quoted sentence denotes that the predictive distribution can capture only the epistemic uncertainty relating to the parameters of sequences, $\theta$, and cannot include the epistemic uncertainty relating to their hyperparameters (if they exist). For example, a sequence model trained using input-output pairs taken from randomly sampled quadratic functions would fail to predict with accuracy the epistemic uncertainty about input-output pairs from a cubic function. Although this appears as model mismatch, if the model is equipped with a polynomial function hyper-prior, a Bayesian inference method can predict accurate epistemic uncertainty. This point was raised to highlight the comparative limitations of epistemic uncertainty of our method in contrast to Bayesian inference approaches. We will add clarity in the manuscript.
>
>
> > There could be several reasonable ways to generate suboptimal data for analysis in toy experiments, ...
>
> We appreciate the insightful suggestion. Indeed, exploring more efficient data collection methods could be beneficial. If the data contains a sufficient proportion of optimal behaviors, the model can learn from these optimal behavior. Included suboptimal data allows the model to manage suboptimal states and learn to improve itself at inference time. A good balance of optimal and suboptimal behaviors can enhance learning efficiency. The exploration of data generation distribution parameters, such as the suggested beta distribution, could lead to more effective learning. We will explore these recommendations in the future.

---

### Meta-Review · Area_Chair_KkjD · 2023-12-10

**Metareview:**

This paper puts forward an approach to address the exploration-exploitation problem in RL via in-context learning, ICEE. They compare to previous methods such as Algorithm Distillation and Decision Transformer, testing in 2 settings: 1) Bayesian optimization and 2) 2D discrete RL environments.

Most reviewers found the paper well-motivated, the experiments were convincing, and the results very promising. Authors’ responses addressed all reviewers’ concerns except for reviewer 1NXH. However I don’t necessarily agree with this reviewer’s response that the approach lacks novelty simply because it relies on importance sampling and Monte Carlo, both of which are very common tools in ML. From my reading, it doesn’t seem as if ICEE is a trivial application of these approaches to ICL. All other reviewers recommended acceptance and I see no reason to reject, although I'm personally of the opinion that it would've been nice to see a wider range of RL tasks included, such as those in the original Decision Transformer and Algorithm Distillation papers.

**Justification For Why Not Higher Score:**

It would have been nice to see evaluation on more environments such as those in the original Decision Transformer and Algorithm Distillation papers.

**Justification For Why Not Lower Score:**

This seems like a promising approach with solid results, and there was a general consensus to accept.

---

### Decision · Program_Chairs · 2024-01-16

Accept (poster)